# Evaluating Bayes Error Estimators on Real-World Datasets with `FeeBee`

**Cedric Renggli**[*]
ETH Zurich

**Luka Rimanic**[*]
ETH Zurich

**Nora Hollenstein**
University of Copenhagen

**Ce Zhang**
ETH Zurich

## Abstract

The Bayes error rate (BER) is a fundamental concept in machine learning that quantifies the best possible accuracy *any* classifier can achieve on a fixed probability distribution. Despite years of research on building estimators of lower and upper bounds for the BER, these were usually compared only on synthetic datasets with known probability distributions, leaving two key questions unanswered: (1) *How well do they perform on realistic, non-synthetic datasets?*, and (2) *How practical are they?* Answering these is not trivial. Apart from the obvious challenge of an *unknown* BER for real-world datasets, there are two main aspects any BER estimator needs to overcome in order to be applicable in real-world settings: (1) the computational and sample complexity, and (2) the sensitivity and selection of hyper-parameters. In this work, we propose `FeeBee`, the first principled framework for analyzing and comparing BER estimators on modern real-world datasets with unknown probability distribution. We achieve this by injecting a controlled amount of label noise and performing multiple evaluations on a series of different noise levels, supported by a theoretical result which allows drawing conclusions about the evolution of the BER. By implementing and analyzing 7 multi-class BER estimators on 6 commonly used datasets of the computer vision and NLP domains, `FeeBee` allows a thorough study of these estimators, clearly identifying strengths and weaknesses of each, whilst being easily deployable on any future BER estimator.

## 1 Introduction

The *Bayes error rate* (BER) [5] is a fundamental concept of machine learning (ML) which quantifies the "irreducible error" of a given task, corresponding to the error rate of a *Bayes optimal classifier*. Given a dataset representative for a task, knowing its exact BER yields the best accuracy any machine learning model could achieve. Being such a fundamental quantity, BER estimators have been applied to a diverse range of applications, from performing feature selection [17, 18], exploring the feature space or behaviour of intermediate representations in trained networks [18], assessing the quality of security defences against ML-based attacks [4], to estimating the feasibility of a ML application given a target accuracy prior to its development [15]. In the *asymptotic regime*, in which one can access an infinite amount of data, one could use a consistent classifier such as k-nearest neighbor (kNN) to estimate the BER [5]. Over the last 50 years, coming up with practical Bayes error estimators in the finite-data regime has been a never ending pursuit of the ML community: from Fukunaga's early effort back in 1975 [10] to Sekeh et al.'s recent effort just last year in 2020 [18] and a diverse collection of other Bayes error estimators [1, 2, 6, 11, 14, 16].

Despite this wide range of real-world applications and the large number of estimators, there exists a technical gap: *BER estimators have only been evaluated and compared on simple synthetic datasets for which the true BER can be calculated*. This results in a lack of understanding of the performance and practicality of BER estimators in real-world scenarios. First, all these synthetic datasets are

---

[*]Equal contribution. Contact: cedric.renggli@inf.ethz.ch or luka.rimanic@inf.ethz.ch.

constructed with simple data generative processes, which can be very different from real-world datasets that are diverse in their modalities (e.g., images, text, tabular data) and data distributions. Second, many BER estimators consist of a hyper-parameter tuning mechanism that relies on the true BERs of synthetic datasets [5, 11, 20], which are in practice not available for real-world datasets. Third, real-world datasets can often take advantage of pre-trained feature transformations which can lead to improving convergence rates at the cost of a slight grow of the BER [16]. Such a behavior is also hard to explore on synthetic datasets.

In this paper, we present FeeBee, which, to our best knowledge, is the first evaluation framework of BER estimators on realistic, non-synthetic datasets. The goal of FeeBee is to enable a systematic study of the performance and practicality of BER estimators in more realistic scenarios. Designing FeeBee is challenging — after all, *how can we compare BER estimators without knowing the true BER?* FeeBee's idea is to go beyond simply evaluating an estimator at a *single point* — instead, evaluate it on a series of points, for which we know the *relative relationship* among their BERs. At its core, FeeBee injects label noise to existing datasets, and using a simple but novel technical result, it estimates the corresponding BER on different noise levels to measure to what degree a BER estimator under/overestimates. The FeeBee framework allows us to systematically evaluate BER estimators. Especially, we focus on two main reasons that hamper the applications of BER estimators in real-world scenarios: **(i)** difficulty of choosing correct hyper-parameters and a feature transformation, and **(ii)** computational and data efficiency. Our key contributions can be summarized as follows:

- We propose FeeBee, the first principled and practical framework for comparing different BER estimators by reasoning about the *evolution* of the true (unknown) BER rather than evaluating the BER as a single value.
- We open-source and deploy FeeBee on 6 well-established, real-world, computer vision and text classification benchmark datasets, on which we systematically evaluate a range of 7 existing BER estimators. The framework can easily be extended with new BER estimators.[2]
- We perform further studies in order to understand the behavior of certain BER estimators, as well as studying the potential of simple (scaled) classifiers to be used as BER estimators.

**Moving forward.** Evaluating BER estimators on real-world datasets is a fundamentally challenging problem. FeeBee provides, to our best knowledge, the first attempt to tackling this problem. We see our contributions as moving away from the synthetic setting by using realistic, well-known datasets for which we are not aware of the true BER. We are aware that this framework is only a step towards covering all the real-world scenarios. Therefore, we hope that FeeBee can help open up future research endeavors towards understanding the behavior of BER estimators on real-world datasets, whilst motivating the development of alternative evaluation frameworks and new BER estimators, followed by further real-world applications of the BER.

## 2 Preliminaries

In this section, we give a short overview over the technical terms and the notation used throughout this paper. Let $\mathcal{X}$ be the feature space and $\mathcal{Y}$ be the label space, with $C = |\mathcal{Y}|$. Let $X, Y$ be random variables taking values in $\mathcal{X}$ and $\mathcal{Y}$, respectively. We denote their joint distribution by $p(X, Y) \sim \mathcal{D}$, often using the simplified notation $p(x, y) = p(X = x, Y = y)$. We define $\eta_y(x) = p(y|x)$ when $C > 2$, and $\eta(x) = p(1|x)$ when $C = 2$, in which case we assume $\mathcal{Y} = \{0, 1\}$.

**Bayes error.** *Bayes optimal classifier* is the classifier that achieves the lowest error rate among all possible classifiers from $\mathcal{X}$ to $\mathcal{Y}$, with respect to $\mathcal{D}$. Its error rate is called the *Bayes error rate (BER)* and we denote it, depending on the context, by $R_{\mathcal{D}}^*$ or $R_{X,Y}^*$, often abbreviated to $R_X^*$ when $Y$ is clear from the context. It can be expressed as

$$R_X^* = \mathbb{E}_X \big[ 1 - \overbrace{\underbrace{\max_{y \in \mathcal{Y}} \underbrace{\eta_y(x)}_{(1)}}_{(3)}}^{(2)} \big]. \tag{1}$$

When examining BER estimators, we only consider methods which are capable of estimating the BER for multi-class classification problems ($C \geq 2$), and divide them into three categories, based on

---

different parts of Equation 1: *density estimators* that estimate (1), *divergence estimators* that address (2), and *estimators* built around the *k-nearest neighbors algorithm*, which focus on (3). In Section 4.1 we distill each of these groups in detail.

**Synthetic vs non-synthetic regimes.** In previous work, even though BER estimators were sometimes applied on real-world datasets (e.g., as a utility for feature reduction strategies, or for quantifying layer-to-layer change in convolutional neural networks, both in [18]), their theoretical properties were tested only on synthetic datasets [5, 6, 11, 12, 18, 20]. Upon knowing the underlying probability distribution, the true BER can then either be computed directly (e.g., [5, 20]), or through a simulation, such as Monte Carlo method [18]. These synthetic datasets often assume Gaussian distributions [11, 12, 20], over small dimensions (at most 8 in [11, 12, 18]). Since a BER estimator is usually constructed based on strong theoretical guarantees in the asymptotic regimes, for each such synthetic dataset one can usually find a set of hyper-parameters which make the estimator predict the true BER reasonably well. However, in order to be able to compare BER estimators, a useful framework needs to give them a chance to be wrong. Therefore, comparing them on synthetic datasets would not yield transferable insights towards real-world, non-synthetic datasets, since they perform well on synthetic datasets, whereas in practice, predicting the true BER is notoriously hard.

## 3 Evaluation Framework

Given a dataset $\mathcal{D}$ containing $n$ i.i.d. samples from $p(X, Y)$, a BER estimator $m$ provides us an estimation of the lower bound $\ell_{\mathcal{D},m}$ and the upper bound $u_{\mathcal{D},m}$ of the *unknown* BER $R_{\mathcal{D}}^*$. In the following, we also assume that we are aware of the state-of-the-art performance (SOTA) of applying machine learning on this dataset: $s_{\mathcal{D}} \geq R_{\mathcal{D}}^* \geq 0$. The goal of our evaluation framework is to come up with some metrics to measure the quality of the lower bound $\ell_{\mathcal{D},m}$ and the upper bound $u_{\mathcal{D},m}$:

$$L_{\mathcal{D}}(m) \in [0, 1] \triangleq \texttt{Sub-optimality of LB estimator } \ell_{\mathcal{D},m}\texttt{; lower the better}$$

$$U_{\mathcal{D}}(m) \in [0, 1] \triangleq \texttt{Sub-optimality of UB estimator } u_{\mathcal{D},m}\texttt{; lower the better}$$

*What is a good evaluation framework?* In this paper, we consider two natural key requirements that we believe need to be satisfied when evaluating BER estimators. First, if the estimator predicts exactly the BER $R_{\mathcal{D}}^*$ for both the lower and the upper bound, then it should hold that $L_{\mathcal{D}}(m) = U_{\mathcal{D}}(m) = 0$. Second, any lower-bound estimate that satisfies $\ell_{\mathcal{D},m} > s_{\mathcal{D}}$ is clearly wrong, whereas any upper-bound estimate that satisfies $u_{\mathcal{D},m} > s_{\mathcal{D}}$ is clearly outperformed by $u = s_{\mathcal{D}}$, SOTA itself. In particular, a good framework should penalize these two cases.

**Challenges of evaluating BER estimators at a single point.** The key challenge of evaluating BER estimators on real-world datasets is that the true BER $R_{\mathcal{D}}^*$ is unknown and we only have access to the SOTA $s_{\mathcal{D}}$. We first show that simply evaluating BER estimators using $s_{\mathcal{D}}$ is challenging in meeting these two natural requirements. The first baseline would use the SOTA as the proxy of $R_{\mathcal{D}}^*$:

$$L_{\mathcal{D}}(m) = |s_{\mathcal{D}} - \ell_{\mathcal{D},m}|, \quad U_{\mathcal{D}}(m) = |u_{\mathcal{D},m} - s_{\mathcal{D}}|.$$

This does not satisfy our first requirement — an "ideal" estimator that always outputs $R_{\mathcal{D}}^*$ would have non-zero sub-optimality as long as $s_{\mathcal{D}} \neq R_{\mathcal{D}}^*$.

We could construct another baseline inspired by the fact that $\ell_{\mathcal{D},m} = 0$ and $u_{\mathcal{D},m} = s_{\mathcal{D}}$ are true bounds and assign them zero sub-optimality:

$$L_{\mathcal{D}}(m) = |\ell_{\mathcal{D},m}|, \quad U_{\mathcal{D}}(m) = |u_{\mathcal{D},m} - s_{\mathcal{D}}|.$$

It satisfies that predicting $\ell_{\mathcal{D},m} = 0$ and $u_{\mathcal{D},m} = s_{\mathcal{D}}$ gives a perfect score. However, the true BER $R_{\mathcal{D}}^*$ has a non-zero score as soon as it differs from $s_{\mathcal{D}}$, again contradicting the first requirement.

**(ii)** One could attempt at penalizing intervals that are *certainly* sub-optimal, for example by

$$L_{\mathcal{D}}(m) = \mathbf{1}_{\{\ell_{\mathcal{D},m} > s_{\mathcal{D}}\}}, \quad U_{\mathcal{D}}(m) = \mathbf{1}_{\{u_{\mathcal{D},m} > s_{\mathcal{D}}\}}.$$

However, this is not distinguishable enough as both the estimator that predicts $\ell_{\mathcal{D},m} = u_{\mathcal{D},m} = 0$ and $\ell_{\mathcal{D},m} = u_{\mathcal{D},m} = s_{\mathcal{D}}$ have perfect score, whereas it is obvious that even though an estimator that always predicts 0 gives a valid lower bound, it is completely non-informative.

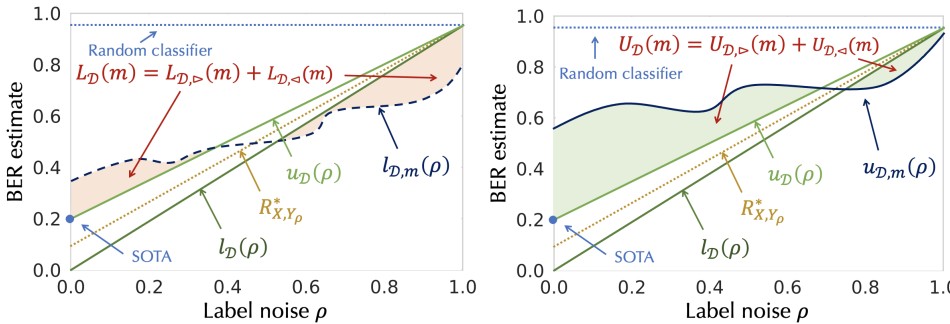

Figure 1: Evaluation Methodology for estimating the lower bound **(left)** and the upper bound **(right)**.

**Evaluating through a series of points.** In order to overcome the above limitation, `FeeBee` injects different levels of label noise and measures how an estimator follows the evolution of the BER, as illustrated in Figure 1. For each sample, with probability $\rho \in [0,1]$ we flip the label, in which case we choose a label uniformly at random. The following result quantifies the increase in the BER.

**Lemma 3.1.** *Let $Y_\rho$ be a random variable defined on $\mathcal{Y}$ by setting $Y_\rho = Z \cdot U(\mathcal{Y}) + (1-Z) \cdot Y$, where $U$ is a uniform variable taking values in $\mathcal{Y}$, and $Z$ is a Bernoulli variable with probability $0 \le \rho \le 1$, both independent of $X$ and $Y$. Then $R^*_{X,Y_\rho} = R^*_{X,Y} + \rho(1 - 1/C - R^*_{X,Y})$.*

PROOF: Let $p(X, Y_\rho)$ be the corresponding joint distribution on $\mathcal{X} \times \mathcal{Y}$ for the random variables $X$ and $Y_\rho$, simplified as $p_\rho(x,y)$. Note that

$$p_\rho(y|x) = \underbrace{p_\rho(y|x, Z=0)}_{p(y|x)} p(Z=0) + \underbrace{p_\rho(y|x, Z=1)}_{p(U=y)} p(Z=1) = (1-\rho)p(y|x) + \frac{\rho}{C}.$$

Thus,
$$R^*_{X,Y_\rho} = \mathbb{E}_X\left[1 - \max_{y \in \mathcal{Y}} p_\rho(y|x)\right] = 1 - \mathbb{E}_X \max_{y \in \mathcal{Y}}\left[(1-\rho)p(y|x) + \rho/C\right]$$
$$= 1 - \rho/C - (1-\rho)\mathbb{E}_X \max_{y \in \mathcal{Y}} p(y|x) = R^*_{X,Y} + \rho(1 - 1/C - R^*_{X,Y}). \qquad \square$$

We remark that $Y_\rho \in \mathcal{Y}$, where $\rho$ corresponds to the probability of randomly changing the original label to a random value in $\mathcal{Y}$ and note that Lemma 3.1 implies that $1 - 1/C \ge R^*_{X,Y_\rho} \ge R^*_{X,Y}$. As a direct consequence of Lemma 3.1, using the SOTA as an upper bound for $R^*_X$, and $R^*_X \ge 0$ as the lower bound, we can define the valid bounds on $R^*_{X,Y_\rho}$:

$$\ell_\mathcal{D}(\rho) = \rho(1 - 1/C), \quad u_\mathcal{D}(\rho) = s_\mathcal{D} + \rho(1 - 1/C - s_\mathcal{D}),$$

yielding $R^*_{X,Y_\rho} \in [\ell_\mathcal{D}(\rho), u_\mathcal{D}(\rho)]$. For a fixed method $m$, we can estimate the *lower bound* $\ell_{\mathcal{D},m}(\rho)$ and the *upper bound* $u_{\mathcal{D},m}(\rho)$ using any BER estimation method on a manipulated dataset $\mathcal{D}_\rho$ obtained by taking $\rho \cdot n$ samples out of $\mathcal{D}$, and randomly changing their labels, whilst keeping the other $(1-\rho) \cdot n$ samples intact. Notice that our evaluation framework needs to change the label on all the data points (i.e. in both the training and test sets). For those estimators that predict only a single value, i.e., try to estimate the exact BER, we set both $\ell_{\mathcal{D},m}(\rho)$ and $u_{\mathcal{D},m}(\rho)$ to that estimate.

As illustrated in Figure 1, in order to define the error of a given method $m$ on the modified dataset $\mathcal{D}_\rho$, we can estimate four areas with respect to the curves: *the area where $m$ clearly under/overestimates the Bayes error lower/upper bound*. More formally, for a given method $m$ we define the *lower BER estimator score* $L_\mathcal{D}(m)$ and the *upper BER estimator score* $U_\mathcal{D}(m)$ by

$$L_\mathcal{D}(m) = L_{\mathcal{D},\triangleright}(m) + L_{\mathcal{D},\triangleleft}(m), \quad U_\mathcal{D}(m) = U_{\mathcal{D},\triangleright}(m) + U_{\mathcal{D},\triangleleft}(m),$$

$$L_{\mathcal{D},\triangleleft}(m) = \frac{2C}{C-1} \int_{\rho=0}^{1} \mathbf{1}_{\{\ell_{\mathcal{D},m}(\rho) < \ell_\mathcal{D}(\rho)\}} (\ell_\mathcal{D}(\rho) - \ell_{\mathcal{D},m}(\rho))d\rho,$$

$$L_{\mathcal{D},\triangleright}(m) = \frac{2C}{C-1} \int_{\rho=0}^{1} \mathbf{1}_{\{\ell_{\mathcal{D},m}(\rho) > u_\mathcal{D}(\rho)\}} (\ell_{\mathcal{D},m}(\rho) - u_\mathcal{D}(\rho))d\rho,$$

$$U_{\mathcal{D},\triangleleft}(m) = \frac{2C}{C-1} \int_{\rho=0}^{1} \mathbf{1}_{\{u_{\mathcal{D},m}(\rho) < \ell_\mathcal{D}(\rho)\}} (\ell_\mathcal{D}(\rho) - u_{\mathcal{D},m}(\rho))d\rho,$$

$$U_{\mathcal{D},\triangleright}(m) = \frac{2C}{C-1} \int_{\rho=0}^{1} \mathbf{1}_{\{u_{\mathcal{D},m}(\rho) > u_\mathcal{D}(\rho)\}} (u_{\mathcal{D},m}(\rho) - u_\mathcal{D}(\rho))d\rho.$$

Table 1: Datasets and state-of-the-art performance on classification tasks. Raw features and thus their dimensionality for NLP tasks do not exist.

| Name | Dimension | Classes $C$ | Train / Test Samples | SOTA % |
|---|---|---|---|---|
| MNIST | 784 | 10 | 60K / 10K | 0.13 [3] |
| CIFAR10 | 3072 | 10 | 50K / 10K | 0.5 [8] |
| CIFAR100 | 3072 | 100 | 50K / 10K | 3.92 [9] |
| IMDB | N/A | 2 | 25K / 25K | 3.79 [21] |
| SST2 | N/A | 2 | 67K / 872 | 3.2 [21] |
| YELP | N/A | 5 | 500K / 50K | 27.80 [21] |

Notation $\triangleright$ and $\triangleleft$ reflects the corresponding upper-left and bottom-right triangles in Figure 1. The scaling constant is chosen in the way that the random classifier, the one that chooses a label uniformly at random, satisfies $L_\mathcal{D}(m) = U_\mathcal{D}(m) = 1$. We will use BER estimator scores $L_\mathcal{D}(m), U_\mathcal{D}(m)$ to assess the performance of existing methods on real-world datasets. We estimate the expectation and standard deviation of the score $S_\mathcal{D}(m)$ by sampling linearly 10 values for $\rho$ and constructing at least 5 random datasets $\mathcal{D}_\rho$ for every sampled $\rho$.

**Discussion.** Going back to the two requirements laid down at the beginning of this section, both are clearly satisfied: **(i)** Lemma 3.1 and the construction of the areas (see Figure 1) prove that predicting $\ell_{\mathcal{D},m}(\rho) = u_{\mathcal{D},m}(\rho) = R^*_{\mathcal{D}_\rho}$ always yields $L_\mathcal{D}(m) = U_\mathcal{D}(m) = 0$. In that case the method $m$ is an *optimal* lower/upper-bound estimate. **(ii)** Estimators that consistently predict $\ell_{\mathcal{D},m}(\rho) = 0$ and $u_{\mathcal{D},m}(\rho) = s_\mathcal{D}$ clearly have $L_\mathcal{D}(m) > 0$ and $U_\mathcal{D}(m) > 0$. Furthermore, for two methods $m$ and $m'$, $\mathbb{E}[L_\mathcal{D}(m)] < \mathbb{E}[L_\mathcal{D}(m')]$ implies that $m$ yields a *better* lower-bound estimate compared to $m'$, whereas $\mathbb{E}[U_\mathcal{D}(m')] > \mathbb{E}[U_\mathcal{D}(m)]$ implies that $m$ yields a *better* upper-bound estimate compared to $m'$, in expectation. This allows one to compare two estimators even when they produce valid bounds.

**Limitations.** The main limitation we see in `FeeBee` is the need of having a relatively strong SOTA value. Note that having no SOTA value would allow even random classifier to satisfy $L_{\mathcal{D},\triangleright}(m) = U_{\mathcal{D},\triangleright}(m) = 0$. In that sense, for `FeeBee` to give valuable insights, we need SOTA to be close to BER to have a wide range of possible values for the areas. We examine this in Section 4.3.

# 4 Analysis of Existing Estimators

In this section, we use our `FeeBee` framework to conduct a systematic study of a diverse range of BER estimators over a collection of real-world datasets.

## 4.1 Setup

**Datasets.** We perform the evaluation on two data modalities that are ubiquitous in modern machine learning — visual classification tasks and text classification tasks, over 6 well-established real-world datasets presented in Table 1. The first group consists of visual classification tasks, including MNIST and CIFAR10/CIFAR100. The second group consists of standard text classification tasks, where we focus on IMDB, SST2, and YELP. Table 1 presents the details of the involved datasets. The splits for all datasets except YELP[3] are taken from Tensorflow Datasets[4]. The SOTA values, especially for the NLP tasks, may sometimes differ as different sources provide values on slightly different splits. A key assumption we made when choosing relevant datasets is that we assume that both the *train* and *test* dataset originate from the same distribution. With that in mind, we rule out datasets for which this is clearly known not to be true, such as ImageNet [13].

**Feature transformations.** BER estimators are usually tested on synthetic data of often small dimension. By testing each estimator on raw data, we observe that on real-world datasets having a transformation that adequately transforms the space and/or reduces the dimension is necessary for every method. We report the results on the raw features (i.e., pixel values) for the vision datasets in Appendix B.1, omitting NLP tasks in this evaluation as there exists no completely *raw* representation. Even though applying a transformation might increase the BER [16], it is often supported by theory, e.g., for kNN-Extrapolate [19] and for kNN [16], by improving convergence. Therefore, we deploy each estimator on a collection of feature transformations, presented in Tables 6 and 7 in Appendix A.

---

[3] https://www.yelp.com/dataset.
[4] https://www.tensorflow.org/datasets/catalog/overview.

Table 2: Overview of BER estimators and their hyper-parameters.

| Estimator | Parameter | Values |
|---|---|---|
| GHP | *None* | |
| DE-kNN,1NN-kNN | k | $[2, 100]$ |
| Gaussian KDE | B | $\{0.0025, 0.05, 0.1, 0.25, 0.5\}$ |
| 1NN / kNN / kNN-LOO/ kNN-Extrapolate | k; dist | $[1, 10]$; $\{cosine, L_2\}$ |

**BER estimators.** The BER estimators implemented initially in our framework can be divided into three groups, motivated by three different parts of Equation 1.

**(1) Density estimators.** Both methods in this group use the full test set with labels to estimate the class posterior, repeated once again for the same test set, but this time without the labels. This allows one to sample the feature space accordingly and get an estimate of the expectation over $\mathcal{X}$. **(DE-kNN)** This method estimates the per-class posterior $\eta_y$ for all $y \in \mathcal{Y}$, by counting the fraction $k_y/k$ of samples with that specific label amongst the $k$ nearest neighbors, in expectation over the feature space [12]. One uses the full test set only to estimate an upper bound by the leave-one-out (LOO) technique, and an optimistic lower bound by the re-substitution technique [11]. **(KDE)** This method estimates the class prior by first taking a fraction of per-class samples in the full test set. Using a kernel density approach, the class likelihood is then estimated using all the samples per class separately. Finally, by using the Bayes formula, one can derive the posterior density per class, which is used as the lower and upper bound.

**(2) Divergence estimator. (GHP)** This estimator uses the generalized Henze-Penrose divergence [18] between every pair of $\eta_i$ and $\eta_j$, to get a provably valid estimator of the BER in the asymptotic regime. The estimation of the divergence can be further utilized to get an upper/lower-bound estimate of the BER. Implementation-wise, using only a single set of samples, one first constructs the minimum spanning tree (MST) over the fully connected graph over all the samples, with edges being defined through Euclidean distances, and then uses the number of dichotomous edges to estimate the BER, noting that GHP and kNN-LOO (introduced below) have similar computational complexity.

**(3) kNN classifier accuracy. (1NN-kNN)** The approach by Devijver [6] aims at estimating the 1NN classifier accuracy by using the k-nearest-neighbor information. In order to get an unbiased estimator of the 1NN classifier accuracy, in [6] it is proposed using $\frac{1}{k(k-1)} \sum_{y \in \mathcal{Y}} k_y(k - k_y)$ as the estimator, where $k$ is the hyper-parameter of the method. This approach is very similar to *DE-kNN*, with the difference that we are not estimating $\eta_y$ based on a test set, but directly the 1NN classifier accuracy. Samples are usually used twice through the resubstitution technique in order to get the 1NN classifier accuracy. **(kNN-Extrapolate)** One major caveat of bounding the BER by any kNN accuracy method lies in the fact that the bounds hold only in the asymptotic regime. As an attempt to surpass this limitation, Snapp and Xu [20] extrapolate the convergence values of kNN for different number of training samples by assuming probability densities with uniformly bounded partial derivatives up through order $N$. However, this requires the number of samples to be exponential in the input dimension and, hence, is challenging to generalize it to representations of higher dimension on real-world datasets. **(1NN)** Inspired by Cover and Hart [5], we define

$$\widehat{R}_{X,1NN} := (R_X)_{n,1}/\Big(1 + \sqrt{1 - \frac{C(R_X)_{n,1}}{C - 1}}\Big), \tag{2}$$

where $(R_X)_{n,k}$ is the validation error of a kNN classifier (with $n$ training samples). The main motivation comes from the fact that in the asymptotic regime, i.e. for $n = \infty$, Cover and Hart [5] proved that the RHS of Equation 2 serves as a lower bound of the BER. **(kNN)** For $k > 1$ and $C > 2$ there is no known bound as strong as for $k = 1$. However, we can still use the same bound even for $k > 1$, even though it is less tight, with a further improvement by Devroye [7] when $C = 2$, yielding

$$\widehat{R}_{X,kNN} = \begin{cases} (R_X)_{n,k}/\big(1 + \sqrt{1 - \frac{C(R_X)_{n,k}}{C-1}}\big), & C > 2, k > 1, \\ (R_X)_{n,k}/(1 + \sqrt{2/k}), & C = 2, k = 2, \\ (R_X)_{n,k}/(1 + \sqrt{1/k}), & C = 2, k > 2. \end{cases} \tag{3}$$

**(kNN-LOO)** When one wants to omit splitting the dataset into test and train sets, the kNN classifier accuracy can be reported using a leave-one-out approach. In this work we simply use this estimator for a fair comparison with certain non-scalable methods, noting that this approach is typically not computationally feasible in practice.

Table 3: $L_{\mathcal{D}}(m)$ **(upper part)** and $U_{\mathcal{D}}(m)$ **(lower part)**: The optimal values per method.

| Dataset | DE-kNN | KDE | GHP | 1NN-kNN | 1NN | kNN | kNN-LOO | kNN_Ext |
|---|---|---|---|---|---|---|---|---|
| MNIST | 0.11 | 0.41 | 0.03 | 0.07 | **0.02** | **0.02** | 0.03 | 0.28 |
| CIFAR10 | 0.14 | 0.36 | 0.07 | 0.10 | 0.05 | **0.03** | **0.03** | 0.34 |
| CIFAR100 | 0.27 | 0.31 | 0.20 | 0.29 | 0.14 | **0.06** | 0.07 | 0.22 |
| IMDB | - | 0.49 | 0.16 | 0.31 | 0.25 | 0.25 | **0.15** | 0.25 |
| SST2 | 0.42 | 0.49 | 0.44 | 0.38 | 0.32 | **0.29** | 0.34 | 0.47 |
| YELP | - | 0.20 | - | - | 0.03 | **0.00** | - | - |
| MNIST | 0.11 | 0.41 | 0.33 | 0.22 | 0.32 | **0.09** | 0.11 | 0.37 |
| CIFAR10 | **0.15** | 0.36 | 0.39 | 0.26 | 0.37 | **0.15** | 0.17 | 0.20 |
| CIFAR100 | 0.28 | 0.31 | 0.55 | 0.36 | 0.49 | 0.31 | 0.37 | **0.21** |
| IMDB | - | 0.49 | 0.51 | 0.39 | 0.60 | 0.43 | 0.34 | **0.28** |
| SST2 | 0.43 | 0.49 | 0.73 | **0.42** | 0.63 | 0.48 | 0.57 | 0.61 |
| YELP | - | **0.20** | - | - | 0.38 | 0.25 | - | - |

**Computational feasibility.** Some BER estimators are not suitable for very large datasets for two reasons: (1) algorithmic or (2) space (i.e., memory) complexity. We restrict the memory and compute time available to run the described evaluation for a single combination of dataset, estimator, transformation and set of hyper-parameters to 45GB and 24h respectively on 4 CPU nodes and access to a single NVIDIA GeForce RTX 2080 Ti GPU. Out of 2376 combinations, 103 ran out of memory and 60 ran out of time (e.g., for high dimensional representations such as BoW, or large datasets such as the test sets of IMDB or YELP).

**Hyper-parameters.** We define a meaningful range of hyper-parameters per method in Table 2. The list was manually curated during the evaluation process by shrinking the range of possible values to get the best possible set of hyper-parameters per dataset and per method. For every combination of dataset and feature transformation, we evaluate each method for every value of the hyper-parameter.

**Experiment protocol.** In order to estimate the quantities described in Section 3, we run every combination of **(1)** dataset, **(2)** pre-trained transformations available for the data modality, **(3)** estimator, and **(4)** the values of their hyper-parameter, multiple times with different random seeds. The code to reproduce all the results along with a public colab that was used to analyze the results are available in the public repository under `https://github.com/DS3Lab/feebee`. We perform 5 independent runs for YELP and 10 independent runs for all other datasets. We sample 11 values linearly between 0.0 and 1.0 for the fraction of label noise $\rho$. We omit reporting derivations of the estimated mean quantities in all tables and plots with more than two lines, as we observe that the variance mostly lies near zero (except for the kNN-Extrapolate method).

### 4.2 Main Results

We now dive into the analysis by examining each presented BER estimator using `FeeBee`, noting that in each plot a solid line represents an upper bound, whereas a dashed line represents a lower bound.

**Analysis of $L_{\mathcal{D}}(m)$ and $U_{\mathcal{D}}(m)$.** The main quantities that `FeeBee` reports are $L_{\mathcal{D}}(m)$ and $U_{\mathcal{D}}(m)$. In Table 3 we list the optimal scores $L_{\mathcal{D}}(m)$ and $U_{\mathcal{D}}(m)$ per method and per dataset, meaning that for each $m$, each dataset and each score, we choose a transformations and hyper-parameters that minimize that score. Further details can be found in Tables 9–20 in Appendix B.2, where we report all the corresponding individual areas $L_{\mathcal{D},\triangleleft}$, $L_{\mathcal{D},\triangleright}$, $U_{\mathcal{D},\triangleleft}$ and $U_{\mathcal{D},\triangleright}$, with further example plots presented in Figure 8 in Appendix B.3.

For $L_{\mathcal{D}}(m)$ we observe that 1NN, kNN, kNN-LOO and GHP are consistently outperforming all the other methods, with either kNN or kNN-LOO being the best choice on each dataset. It is important to note that the main contribution to $L_{\mathcal{D}}(m)$ in the case of 1NN, kNN, kNN-LOO and GHP comes from $L_{\mathcal{D},\triangleright}(m)$, whereas for DE-kNN, KDE and kNN-Extrapolate there is significant contribution from $L_{\mathcal{D},\triangleleft}(m)$. This yields that the first group will only get better as more transformations become available (by reducing the bias that a transformation introduces), whereas the second group provides less informative lower-bound estimators. For $U_{\mathcal{D}}(m)$, we see that there is no method that consistently outperforms the others. Out of the well-performing methods on $L_{\mathcal{D}}(m)$, 1NN and GHP are tangibly inferior to kNN and kNN-LOO, noting similarly as above that the main contribution for these methods comes from $U_{\mathcal{D},\triangleright}(m)$ which will decrease with better feature transformations.

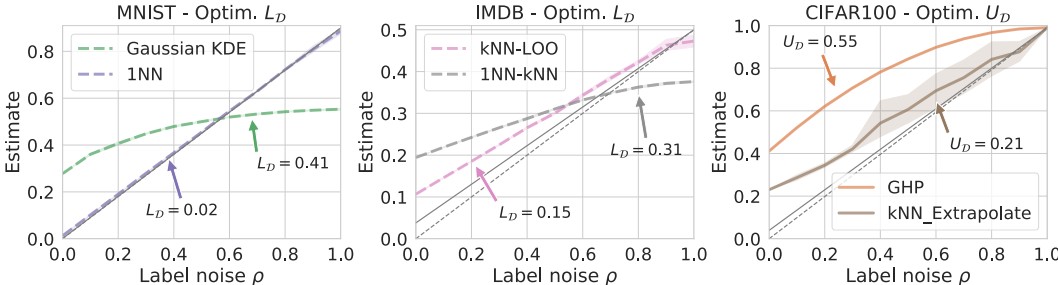

Figure 2: Plotting two methods where one of them has low and the other has high **(left, middle)** $L_{\mathcal{D}}(m)$, or **(right)** $U_{\mathcal{D}}(m)$, confirming that numerical values correspond to estimator's quality. The shaded area represents the 95% quantiles.

Table 4: $\ell_{\mathcal{D},m}(0)$ vs $L_{\mathcal{D}}(m)$: The difference between the *optimal* $L_{\mathcal{D}}(m)$ and the $L_{\mathcal{D}}(m)$ that is calculated for hyper-parameters and transformations that minimize $\ell_{\mathcal{D},m}(0)$.

| Dataset | DE-kNN | KDE | GHP | 1NN-kNN | 1NN | $k_{>1}$NN | kNN-LOO | kNN_Ext |
|---|---|---|---|---|---|---|---|---|
| MNIST | 0.24 | 0.59 | **0.00** | 0.56 | **0.00** | 0.06 | 0.04 | 0.37 |
| CIFAR10 | 0.21 | 0.64 | **0.00** | 0.52 | **0.00** | 0.11 | 0.10 | 0.42 |
| CIFAR100 | 0.10 | 0.69 | **0.00** | 0.30 | **0.00** | 0.05 | 0.02 | 0.59 |
| IMDB | - | 0.01 | **0.00** | 0.26 | 0.01 | 0.14 | 0.20 | 0.33 |
| SST2 | 0.00 | 0.50 | 0.01 | 0.20 | **0.00** | 0.11 | 0.09 | 0.30 |
| YELP | - | 0.09 | - | - | **0.00** | **0.00** | - | - |

We also note that BER estimators are often performing better on certain type of feature transformations, as seen in Tables 9–20 in the supplementary materials. E.g., 1NN and kNN perform better on pre-trained embeddings (supporting [16]), whereas kNN-Extrapolate performs best under transformations that reduce the dimension, such as a low-dimensional PCA (supporting [19]).

**Numerical values and estimator's quality.** A key question is whether the numerical quantities correspond to the performance: *does a lower $L_{\mathcal{D}}(m)$ imply a better lower-bound estimator?* For simplicity, in Figure 2 we plot over 3 different dataset, 2 methods each such that one has a high score and the other has a low score and see that their ability to follow the evolution of the BER differs. We provide full graphs of each methods over all datasets in Appendix B.2, positively answering the above question.

**Minimizing $\ell_{\mathcal{D},m}(0)$ vs minimizing $L_{\mathcal{D}}(m)$.** As described in Section 3, whilst an estimator that always predicts zero gives a valid lower bound, its behavior is non-informative. However, for some methods tuning for $\ell_{\mathcal{D},m}(0)$ might be (close to) optimal for $L_{\mathcal{D}}(m)$. Thus, in Table 4, for each method and over each dataset we report the difference in the optimal $L_{\mathcal{D}}(m)$, i.e. the best hyper-parameters and the best transformation, and the one chosen by hyper-parameters and transformation that minimize $\ell_{\mathcal{D},m}(0)$. We observe that 1NN and GHP are the only ones that are fully robust in the sense that choosing the best hyper-parameters and transformation for $\ell_{\mathcal{D},m}(0)$ yields close to optimal $L_{\mathcal{D}}(m)$, whereas for every other method we can find examples in which the choice based on $\ell_{\mathcal{D},m}(0)$ is significantly suboptimal for $L_{\mathcal{D}}(m)$.

### 4.3 Further Discussion

**Influence of SOTA.** In general, we observe that our framework works when the SOTA value is not too weak. This can be seen by comparing the results on YELP (SOTA error of 27.80%) vs other datasets (SOTA errors of at most 3.92%). For example, on YELP kNN has $L_{\mathcal{D}}(m) = 0.0$ (see Table 3), whilst clearly having difficulties in following the BER evolution (see Figure 7a in Appendix B.2, top-right). A stronger SOTA would detect such difficulties for kNN on YELP through a non-zero $L_{\mathcal{D}}(m)$. Other datasets involved in this framework satisfy that these have been intensively studied by the research community, particularly in the last few years, resulting in strong SOTA values.

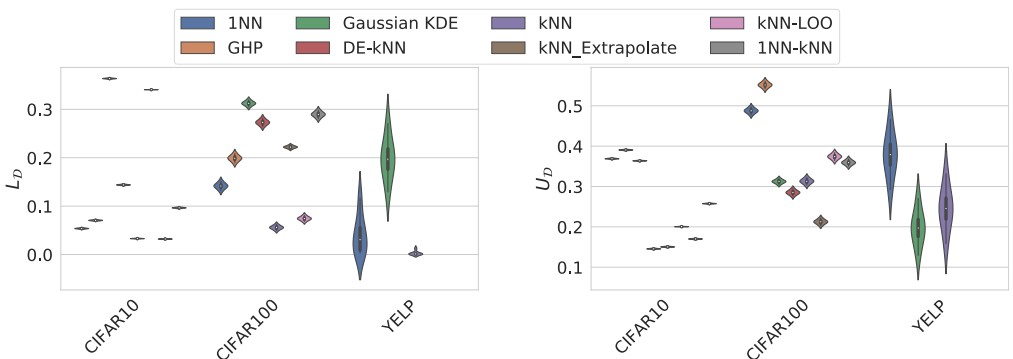

Figure 3: Sensitivity of `FeeBee` with respect to changes in the SOTA values for **(left)** $L_{\mathcal{D}}(m)$, and **(right)** $U_{\mathcal{D}}(m)$. Strong SOTA values are robust to changes (CIFAR10, CIFAR100), whereas weak SOTA values suffer from greater uncertainty (YELP).

**Sensitivity to changes in SOTA.**   In order to study `FeeBee`'s robustness with respect to changes in SOTA values, we perform additional experiments by altering SOTA. In Figure 3 we show three examples in which SOTA values inserted into the framework are SOTA $\pm$ $\delta\cdot$ SOTA, for $\delta \in \{0, 0.05, 0.1, 0.25\}$, which, we believe, represent realistic improvements in the near future. On datasets which have strong SOTAs (all but YELP), we see that modifying the values has almost no impact on `FeeBee`'s insights (kNN and kNN-LOO perform best on CIFAR10 and CIFAR100, followed by 1NN and GHP, cf. Table 3). By comparing CIFAR10 and CIFAR100 we see that the robustness is further improved with stronger SOTA values. Therefore, we conclude that `FeeBee` is rather robust to small changes in the values when SOTA is relatively strong. On the other hand, when the SOTA value is weak, as in the case of YELP, Figure 3 implies that the sensitive of SOTA value starts to increase, introducing significantly more uncertainty than in the other datasets. However, we remark that the insights are still useful and preserved even in this difficult case.

**Alternative flipping strategies.**   One could create noisy labels in many different ways. For example, by assigning different flipping probabilities to different classes, or by conditioning on certain features and flipping differently on these features (e.g., different probabilities for day vs night images). In most of these cases one could produce an analogue of Lemma 3.1 which would provide foundations for an alternative framework. However, each such method would put certain constraints on the applicable datasets which makes the task of constructing a framework inherently more difficult. Having in mind that `FeeBee` in this simplest form is already successful in distinguishing existing estimators, and our belief that it will stay successful in the future, we opt for using the simplest such framework.

**GHP vs 1NN.**   When looking at the derivation of both the upper and lower bounds of GHP (Theorem 1 in [18]), we see that, asymptotically, this method averages the number of dichotomous edges connecting a sample from two different classes, on the minimum spanning tree (MST) that is connecting all the samples in Euclidean space. The 1NN-LOO estimator implicitly performs the same task over the 1-nearest-neighbor graph, instead of the MST. Intuitively, the 1-nearest-neighbor graph should lead to a slightly lower error rate when compared to the MST. In particular, 1NN-LOO should outperform GHP if the feature transformation forms well-separable clusters of samples from the same class. Nevertheless, showing this is beyond the scope of this work and left for future work.

**kNN estimator for** $k > 1$**.**   Due to its success in minimizing $L_{\mathcal{D}}(m)$, whilst being inferior to 1NN and GHP in choosing hyper-parameters and transformations based on $\ell_{\mathcal{D},m}(0)$, we further examine the kNN estimator. In Figure 4a we plot 1NN and kNN for $k \in \{3, 5, 7, 9\}$ on two representing datasets and over transformations that are minimizing $L_{\mathcal{D}}(m)$, with corresponding values in Table 5. We see that increasing $k$ increases $L_{\mathcal{D},\triangleleft}(m)$, which makes kNN a worse lower-bound BER estimator since it provides a less informative lower bound. We believe that the reason for such a behavior lies in the fact that for $k > 1$ the only relevant bounds are given in the case when $C = 2$, by Devroye [7], whereas for $C > 2$ and $k > 1$ there are no known bounds as strong as the one of Cover and Hart [5] for $k = 1$. This results in a larger gap between the BER and the lower bound in the asymptotic regime. In the finite regime, at the moment this is mitigated by the bias that feature transformations introduce, however, in the future we expect this bias to decrease and, thus, kNN might further suffer from these loose bounds.

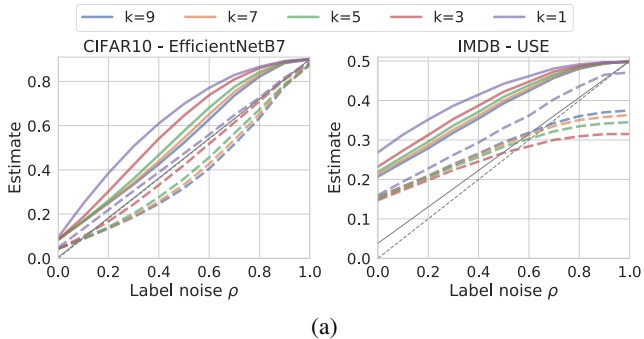
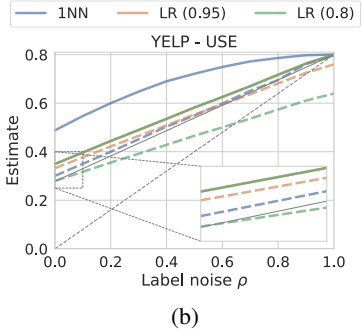

| | (a) | | (b) |

Figure 4: **(a)** kNN estimator for $k > 1$: Increasing $k$ increases $L_{\mathcal{D},\lhd}(kNN)$. **(b)** Logistic regression (LR) as BER estimator: Larger scaling decreases $U_{\mathcal{D}}(m)$ further, but increases $L_{\mathcal{D},\lhd}(m)$.

Table 5: Impact of $k > 1$ and LR Model with different constants vs 1NN.

| Dataset | Method | Transformation | $U_{\mathcal{D},\rhd}(m)$ | $L_{\mathcal{D}}(m)$ | $L_{\mathcal{D},\rhd}(m)$ | $L_{\mathcal{D},\lhd}(m)$ |
|---------|--------|----------------|------|------|------|------|
| CIFAR10 | 1NN | EfficientNet-B7 | 0.37 | 0.05 | 0.05 | 0.00 |
| | 3NN | EfficientNet-B7 | 0.28 | 0.04 | 0.01 | 0.03 |
| | 5NN | EfficientNet-B7 | 0.21 | 0.11 | 0.01 | 0.11 |
| | 7NN | EfficientNet-B7 | 0.18 | 0.14 | 0.01 | 0.14 |
| | 9NN | EfficientNet-B7 | 0.16 | 0.16 | 0.01 | 0.15 |
| IMDB | 1NN | USE | 0.60 | 0.26 | 0.25 | 0.01 |
| | 3NN | USE | 0.52 | 0.30 | 0.12 | 0.17 |
| | 5NN | USE | 0.48 | 0.27 | 0.14 | 0.13 |
| | 7NN | USE | 0.45 | 0.26 | 0.15 | 0.11 |
| | 9NN | USE | 0.43 | 0.25 | 0.16 | 0.09 |
| YELP | 1NN | USE | 0.38 | 0.03 | 0.03 | 0.00 |
| | LR Model (0.8) | USE | 0.10 | 0.08 | 0.00 | 0.08 |
| | LR Model (0.95) | USE | 0.10 | 0.06 | 0.05 | 0.01 |

**Improved upper bounds.** In Table 3, we see that BER estimators are better at minimizing $L_{\mathcal{D}}(m)$ than $U_{\mathcal{D}}(m)$. Even though any classifier can be used as an upper-bound BER estimator, to the best of our knowledge, the estimators 1NN and kNN from Equations 2 and 3 are the only classifier-based estimators for which one has theoretical guarantees ([5] for 1NN and [7] for kNN) for the lower bound on the BER. However, one can construct a lower-bound estimator by scaling the accuracy of the estimator by some constant $c \in (0,1)$. We test the simplest such estimator – logistic regression (LR) on top of the frozen representations, and several scaling constants. In Figure 4b we observe that even though LR yields a significantly better estimator of the upper bound than 1NN, it has worse $L_{\mathcal{D}}(m)$ even for the best scaling constant. Furthermore, we note that increasing the scaling, in order to reduce $L_{\mathcal{D},\rhd}(m)$, further increases $L_{\mathcal{D},\lhd}(m)$, yielding a less informative lower-bound estimator, also visible in Table 5.

## 5 Conclusion

In this work, we introduced `FeeBee`, the first system to systematically evaluate lower and upper bound estimators of the BER on real-world data. By providing a thorough analysis of existing estimators using `FeeBee`, we observe that GHP and 1NN are consistently outperforming the other lower bound estimators whilst enabling easy hyper-parameter selection on a single point estimate. We also observe that currently there does not exist an estimator that is able to simultaneously outperform other estimators on both lower and upper bound. We are open-sourcing the framework which is easily extendable with any new BER estimator, and hope to further motivate and enable progress in building better and more practical methods in the future.

## Acknowledgements

CZ and the DS3Lab gratefully acknowledge the support from the Swiss National Science Foundation (Project Number 200021_184628, and 197485), Innosuisse/SNF BRIDGE Discovery (Project Number 40B2-0_187132), European Union Horizon 2020 Research and Innovation Programme (DAPHNE, 957407), Botnar Research Centre for Child Health, Swiss Data Science Center, Alibaba, Cisco, eBay, Google Focused Research Awards, Kuaishou Inc., Oracle Labs, Zurich Insurance, and the Department of Computer Science at ETH Zurich.

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
