# OpenReview forum: "Evaluating Bayes Error Estimators on Real-World Datasets with FeeBee"
_NeurIPS.cc/2021/Track/Datasets_and_Benchmarks/Round2 — NeurIPS 2021 Datasets and Benchmarks Track (Round 2)_

### Official Review · Reviewer_rwUd · 2021-09-06
**A well-written, detailed set of benchmarks to evaluate BER estimators on a range of standard ML datasets**

**Rating:** 7
**Confidence:** 2

**Strengths:**

To the reviewer’s best knowledge, the framework is the first of its kind for evaluating BER without the presence of a known probability distribution. The fundamental set of assumptions and theory used to construct lower and upper bounds seem sound. The paper conducts very thorough experiments to support the author’s hypotheses. Importantly, all the experiments are performed transparently with a Google colab notebook, which ensures reproducibility.

Additionally, the paper is well written and manages to introduce technical content without being overwhelming to unfamiliar readers (such as myself).

An additional strength is the scope and applicability of this work, the majority of the NeurIPS community is likely to find this work relevant as it involves benchmark datasets commonly encountered in the literature.


**Weaknesses:**

The main limitation of this work concerns the statement:

> A key assumption we made when choosing relevant datasets is that we assume that both the train and test dataset originate from the same distribution. With that in mind, we rule out datasets for which this is clearly known not to be true, such as ImageNet [13].

This paper is intending to address the question:
>  (1) How well do they perform on real-world datasets?

Yet to the best of my knowledge, unfortunately real-world datasets rarely conform to those assumptions. Furthermore, all experiments presented in this paper are limited to MNIST, CIFAR, IMDB, SST2 and YELP, and as such I am not sure about how much this paper does to advance BER estimation on *real-world* data. I still however think it is a reasonable choice as a first step, and perhaps I would only change the wording of the abstract and contributions to reflect this.

Furthermore, this method relies on the presence of a SOTA value, which once again for researchers working on their own problems will not be known in advance.


**Additional Feedback:**

Please engage with my review in the rebuttal to help clarify a few points and I will update my score in accordance with other reviewers and your responses. Thank you for your understanding.

Edit: Based on author's responses and further clarifications, I have adjusted my score to 7.

**Clarity:**

The paper is very well written and relatively easy to follow even for those unfamiliar to the topic. Mathematical notation is well used to define quantities. As a few minor points I would like to state:

1. I would like to see figures with $U_\mathcal{D}(m)$ and $L_\mathcal{D}(m)$ on the same graphs, as at a glance it can reveal problems when the two cross over (e.g. kNN_Ext CIFAR10) and to more easily visualise the bounds between the two quantities.
2. Please upgrade the format of the figures to be of `pdf` or other vector graphic format, so that for any DPI and zoom level all the lines retain the best clarity. Additionally, it is difficult to differentiate the colour scheme (e.g. Fig 3a, 1NN, DE-kNN, and kNN-LOO of Appendix Fig 4)

**Correctness:**

Other than my note in weaknesses, the paper appears correct: benchmarks were chosen reasonably within the set of assumptions, and experiments were conducted in detail. The appendix also supplies more detail to help understand the reasoning presented in the paper. I will defer to the opinion of reviewers more knowledgeable in this field than myself to assess my final score post rebuttal.



**Documentation:**

Overall there appears to be sufficient detail to support reproducibility. To ensure access remains consistent over long periods of time and reproducible, I would advise the colab (which visualises results etc) to be present in the main repository and remove some of the dependency on GDrive as an option. The documentation in the main code repository looks like a very nice start, but it could benefit from separate tutorials linked on e.g. the implementation of other BER methods, to expand on the main readme.

**Ethics:**

To the best of my knowledge there are no ethical concerns to discuss with this submission.

**Relation To Prior Work:**

There is no dedicated section on its relation to prior work, and while authors state that there is no other framework to evaluate BER on real-world data, a section between 1. and 2. could be added to describe applications on synthetic datasets for the reader to understand the differences in evaluation of the methods (between synthetic and real: in the presence of a prob. distribution vs in the absence). With the extra page limit afforded during the rebuttal phase, this could be a useful addition to further strengthen the work.

**Summary And Contributions:**

This paper is categorised as a benchmark paper for the Dataset and Benchmark track. I would like to preface this review with a statement that this paper is outside of the area of my expertise, so if any of my comments show trivial errors of judgement, please engage in rebuttals constructively!

The paper proposes the `FeeBee` framework to quantify the Bayes Error Rate (BER) for real-world datasets with no known probability distribution. The authors demonstrate this by injecting a controlled amount of label noise and performing multiple evaluations on a series of different noise levels. The authors implement and analyse a range of multi-class BER estimators on 6 popular ML benchmarks to demonstrate their framework. They conclude that GHP and 1NN are consistently outperforming other lower bound estimators whilst enabling easy hyper-parameter selection on a single point estimate.

The main contribution is the open-source framework `FeeBee`, the first system to systematically evaluate lower and upper bound estimators of the BER on real-world data under a set of assumptions. The framework is built to motivate further research in new BER estimators to enable progress in building better and more practical methods in the future.

---

> ### Author Response · Authors · 2021-09-26
> **Abstract, Clarity, Prior work, Documentation**
>
> We thank the reviewer for a careful and thorough analysis of our work, and the willingness to engage in a discussion. In the next few days we will use the additional page to incorporate suggested changes, which we outline here.
>
>
> >The main limitation of this work concerns the statement:
> A key assumption we made when choosing relevant datasets is that we assume that both the train and test dataset originate from the same distribution...
>
> ...
>
> >I still however think it is a reasonable choice as a first step, and perhaps I would only change the wording of the abstract and contributions to reflect this.
> Furthermore, this method relies on the presence of a SOTA value, which once again for researchers working on their own problems will not be known in advance.
>
> We agree that the abstract might unintentionally lead to the expectation that our framework would easily generalize to any real-world dataset. We should have scoped this much better -- we see our contributions as moving away from the synthetic setting by using realistic, well-known datasets for which we are not aware of the true Bayes error. We agree that this is far from covering all the “real-world” scenarios --- To reflect this, we changed the wording in the abstract (by placing "realistic, non-synthetic" instead of "real-world", and removing "any" in "any modern real-world") and reflected this in the main body, thank you for this valuable comment. -- Please do let us know if there are additional things we can do to precisely scope our work.
>
> Moreover, we will also add an additional discussion paragraph about these limitations, hoping that it can inspire future research.
>
> > As a few minor points I would like to state:
> I would like to see figures with UD(m) and LD(m) on the same graphs, as at a glance it can reveal problems when the two cross over (e.g. kNN_Ext CIFAR10) and to more easily visualise the bounds between the two quantities.
> Please upgrade the format of the figures to be of pdf or other vector graphic format, so that for any DPI and zoom level all the lines retain the best clarity. Additionally, it is difficult to differentiate the colour scheme (e.g. Fig 3a, 1NN, DE-kNN, and kNN-LOO of Appendix Fig 4)
>
> 1) Figures for $U_D(m)$ and $L_D(m)$ are not comparable per se because when we denote "Optimal $U_D(m)$" and "Optimal $L_D(m)$", this means that we minimize for $U_D(m)$ and $L_D(m)$, respectively. This (often) yields different hyper-parameters and, more importantly, different transformations, as seen in Tables 9-20. We remark that sometimes we do report both upper and lower bound over a fixed transformation (e.g., in Figure 3, for kNN in (a), and logistic regression in (b), where the transformation is written in the title) and we thought about adding figures with more estimators. However, these are often too crowded so we avoided them unless there was a particular message to be sent (as in Figure 3).
> Having said that, we will attach some of these results in the supplementary materials, also showing that there are no crossovers under the same set of hyper-parameters and transformations.
> 2) We updated all our figures to .pdf and improved the clarity, thank you for that suggestion. We tried several different color schemes and this seemed to be the best one, but we will fine-tune the color scheme for the camera-ready version once we agree on all the plots.
>
> > There is no dedicated section on its relation to prior work, and while authors state that there is no other framework to evaluate BER on real-world data, a section between 1. and 2. could be added to describe applications on synthetic datasets for the reader to understand the differences in evaluation of the methods (between synthetic and real: in the presence of a prob. distribution vs in the absence)...
>
> We thank the reviewer for the suggestion and we will add such a section. We believe that the best place for it is in Section 2, after defining the BER and as a motivation Section 3.
>
> >Overall there appears to be sufficient detail to support reproducibility. To ensure access remains consistent over long periods of time and reproducible...
>
> We agree with everything that is written and we plan to spend further time on the documentation and accessibility for the camera-ready version.

---

> > ### Author Response · Authors · 2021-09-28
> > **Updated paper**
> >
> > We want to inform the reviewer that we updated our paper and used the additional page to add several new paragraphs based on reviewer's comment, as explained in our previous answer:
> > - We changed the abstract and in "Moving forward" paragraph in the introduction we added a discussion on "real-world" vs "realistic, non-synthetic"
> > - We added "Synthetic vs non-synthetic regimes" paragraph at the end of Section 2
> > - We added an additional section in the appendices: "B.3 Further Example Plots", in which we plot samples of FeeBee's output, where one can also see that there are no crossings for kNN-Extrapolate when $L_{\mathcal{D}}(m)$ and $U_{\mathcal{D}}(m)$ are comparable (under the same transformation and hyperparameters)
> >
> > We are grateful to the reviewer for raising several valuable questions which definitely improved our paper. We will gladly discuss any further concerns.

---

> > ### Comment · Reviewer_rwUd · 2021-09-28
> > **Thank you for addressing the comments.**
> >
> > I'd like to thank authors for accepting suggestions: **I agree to all the changes outlined by the authors**. As a final minor point, I would like point (1.) to be made clear in the paper, though it may have been done so already and was misunderstood by me (I am not very familiar with this area). With these changes in place I would be happy to raise my score to 7.

---

### Official Review · Reviewer_kzad · 2021-09-21
**Interesting paper**

**Rating:** 7
**Confidence:** 2
**Correctness:** Evaluations and claims are correct to…
**Clarity:** Yes

**Strengths:**

- Strong and extensive evaluation. Deployed on 6 datasets of text and image, evaluation on 7 BER estimators.
- Good analysis of various BER estimation methods.

**Weaknesses:**

- in the terms of methodology flipping labels at random may not be the best way to create noisy labels. This is not very realistic comparing to human raters. In reality disparity of vote among raters indicates difficulty of a specific instance which is not taken into account here.


**Additional Feedback:**

All has been mentioned above.

**Documentation:**

The paper is not about introducing a new method. Everything about the evaluation setup is specified and source code is available.

**Ethics:**

I did not find ethical concerns.

**Relation To Prior Work:**

Yes

**Summary And Contributions:**

The paper presents a practical framework for evaluation of base error rate estimators on real world datasets. Instead of evaluating BER on a single point, feebee, uses the relative relationship among BERs on a series of points. This is done by injecting a controlled amount of label noise to datasets and evaluating how BER changes i.e. under estimates or over estimates the noise amount.

---

> ### Author Response · Authors · 2021-09-26
> **Alternative flipping strategies**
>
> > in the terms of methodology flipping labels at random may not be the best way to create noisy labels. This is not very realistic comparing to human raters. In reality disparity of vote among raters indicates difficulty of a specific instance which is not taken into account here.
>
> We thank the reviewer for raising a very interesting point. We agree that there are many ways of creating noisy labels that take these things into account, for example by assigning different flipping probabilities to different clusters, or by conditioning on certain features and flipping differently on these features (for example, having different probabilities for day vs night images). In many of these cases we believe that one could produce an analogue of Lemma 3.1 which would provide foundations for an alternative framework -- this is very interesting future work and we will include a paragraph to summarize our thinking here ("Alternative flipping strategies") in Section 4.3 and we thank the reviewer for this important insight.

---

> > ### Author Response · Authors · 2021-09-28
> > **Updated paper**
> >
> > We want to inform the reviewer that we have updated our paper and used the additional page to discuss alternative flipping strategies (added to Section 4.3 as the paragraph "Alternative flipping strategies").
> >
> > We thank the reviewer for raising this important question which improved our exposition.

---

### Official Review · Reviewer_SyzK · 2021-09-22

**Rating:** 6
**Confidence:** 2
**Correctness:** The paper is correct to my knowledge.
**Clarity:** The paper is written clearly.

**Strengths:**

- The paper proposes a novel framework for evaluating BER estimators without knowing the data distribution or the ground-truth BER values, based on a simple, but novel technical result. This addresses the key limitation of prior evaluation methods, and allows for evaluation on real-world datasets
- The paper evaluates a thorough benchmark of 7 BER estimators on 6 datasets (text and images)
- The paper is written clearly

**Weaknesses:**

- At the end of section 3, the authors say that FeeBee needs a strong SOTA value. How strong of a SOTA value is necessary, and how sensitive are the results to this?

**Additional Feedback:**

Typos
- Line 86 "unkonwn"

**Documentation:**

Yes, the paper includes an extensive description of the benchmark setup in section 4.1, and the authors include a repository that includes the code to reproduce all results.

**Ethics:**

Not to my knowledge.

**Relation To Prior Work:**

The contribution wrt the prior work is clear. Unlike existing evaluation of BER estimators, FeeBee doesn't rely on knowing the ground-truth BER or the data distribution, and this allows FeeBee to be applied on real-world datasets, not just synthetic ones. FeeBee estimates BER by injecting different amounts of label noise into the datasets and applying their technical result from section 3 to derive an estimate.

**Summary And Contributions:**

- The authors propose FeeBee, which is an evaluation framework for bayes error risk (BER) estimators. FeeBee can be used to benchmark BER estimators on real-world dataset because it doesn't rely on knowing the ground-truth BER and data distribution unlike previous evaluation frameworks
- The authors apply FeeBee to benchmark 7 BER estimators on 6 real-world datasets

---

> ### Author Response · Authors · 2021-09-26
> **A discussion on SOTA values**
>
> > At the end of section 3, the authors say that FeeBee needs a strong SOTA value. How strong of a SOTA value is necessary, and how sensitive are the results to this?
>
> We thank the reviewer for pointing this out! We plan to address this in two ways.
>
> First, we plan to include a sensitivity study about SOTA values  -- Our current plan is to test small intervals around SOTA, say SOTA  $\pm \ \delta\cdot$ SOTA, for $\delta \in \{0.05,0.1,0.25\}$, by reporting optimal $L_{\mathcal{D}}(m)$ and $U_{\mathcal{D}}(m)$. We expect the SOTA values to be robust on small changes and have little to no influence on the ranking of the methods. If you have any feedback on the protocol, please let us know.
>
> In addition to this sensitivity study, we also plan to add an additional paragraph to discuss the impact of SOTA across datasets (Section 4.3), noting the following. The influence of having a strong SOTA can be seen in YELP (SOTA accuracy of 72.20%) vs other datasets (SOTA accuracies of at least 96.08%). For example, on YELP kNN has $L_{\mathcal{D}}(m)=0.0$ (Table 3), whilst clearly having difficulties in following the BER evolution (Figure 6, top-right). Better SOTA would be able to detect that kNN has difficulties on YELP through a non-zero $L_{\mathcal{D}}(m)$. This is one of the reasons why we perform an additional study on kNN in Section 4.3.
>
> We will provide the results and update the manuscript in the next few days. Thank you for reporting the typo.

---

> > ### Author Response · Authors · 2021-09-28
> > **Updated manuscript**
> >
> > We want to inform the reviewer that we updated our manuscript and used the additional page to study the influence of SOTA values on our framework, as explained in our previous comment.
> >
> > We added Figure 3 and two additional paragraphs in Section 4.3:
> > - "Influence of SOTA"
> > - "Sensitivity to changes in SOTA"
> >
> > The results are performing as expected --- In Figure 3 we see that our evaluation framework can work as long as SOTA is not too weak, since adding perturbations on most SOTA values lead to robust insights. On the other hand, if the SOTA is weak (e.g., YELP), the sensitivity starts to increase.
> >
> > We appreciate the reviewer for raising this important question and believe that this addition makes the paper much stronger.

---

### Decision · Program_Chairs · 2021-10-09

**Decision:**

Accept

**Comment:**

All reviewers agree on acceptance. I recommend the authors to take into account the reviewers' comments to improve the paper for its camera-ready version.